# Proton Conductivity of La$_2$(Hf$_{2-x}$La$_x$)O$_{7-x/2}$ "Stuffed" Pyrochlores

Anna V. Shlyakhtina [1,2,*], Nikolay V. Lyskov [3,4], Galina E. Nikiforova [2], Anna V. Kasyanova [5,6], Galina A. Vorobieva [1], Igor V. Kolbanev [1], Dmitry N. Stolbov [7] and Dmitry A. Medvedev [5,6]

[1] N.N. Semenov Federal Research Center for Chemical Physics, Russian Academy of Sciences, 119991 Moscow, Russia; vorob@chph.ras.ru (G.A.V.); annash@chph.ras.ru (I.V.K.)
[2] Kurnakov Institute of General and Inorganic Chemistry of the Russian Academy of Sciences, 119991 Moscow, Russia; gen@igic.ras.ru
[3] Institute of Problems of Chemical Physics RAS, Moscow Region, 142432 Chernogolovka, Russia; lyskov@icp.ac.ru
[4] Department of Physics, HSE University, Myasnitskaya str. 20, 101000 Moscow, Russia
[5] Institute of High Temperature Electrochemistry, Ural Branch, Russian Academy of Sciences, 620219 Ekaterinburg, Russia; kasyanova.1996@list.ru (A.V.K.); dmitrymedv@mail.ru (D.A.M.)
[6] Hydrogen Energy Laboratory, Ural Federal University, 620002 Ekaterinburg, Russia
[7] Department of Chemistry, Lomonosov Moscow State University, Leninskie Gory 1-3, 119991 Moscow, Russia; stolbovdn@gmail.com
* Correspondence: annashl@inbox.ru or annashlyakhtina61@gmail.com; Tel.: +7-(495)939-74-50

**Featured Application: The proposed materials can be applied in solid oxide fuel cells (SOFCs).**

**Abstract:** The design of new oxygen- and proton-conducting materials is of paramount importance for their possible utilization in solid oxide fuel cells. In the present work, La$_2$(Hf$_{2-x}$La$_x$)O$_{7-x/2}$ ($x = 0, 0.1$) ceramics were prepared using ball milling of oxide mixtures (La$_2$O$_3$ and HfO$_2$) followed by high-temperature annealing at 1600 °C for 10 h in air. La$_2$Hf$_2$O$_7$ ceramics exhibit an ordered pyrochlore-type structure, whereas La$_2$(Hf$_{1.9}$La$_{0.1}$)O$_{6.95}$ has a defect pyrochlore structure type with oxygen vacancies at the 48f positions. The oxygen ion and proton conductivity of La$_2$(Hf$_{1.9}$La$_{0.1}$)O$_{6.95}$ "stuffed" pyrochlore ceramics was investigated by electrochemical impedance spectroscopy (two-probe AC) and four-probe DC measurements in a dry and a wet atmosphere (air and nitrogen). The use of two distinct conductivity measurement techniques ensured, for the first time, the collection of reliable data on the proton conductivity of the La$_2$(Hf$_{1.9}$La$_{0.1}$)O$_{6.95}$ "stuffed" hafnate pyrochlore. La$_2$Hf$_2$O$_7$ was found to be a dielectric in the range 400–900 °C, whereas the La$_2$(Hf$_{1.9}$La$_{0.1}$)O$_{6.95}$ "stuffed" pyrochlore had both oxygen ion and proton conductivities in this temperature range. The proton conductivity level was found to be equal to ~8 × 10$^{-5}$ S/cm at 700 °C. Clearly, the proton conductivity of the La$_2$(Hf$_{1.9}$La$_{0.1}$)O$_{6.95}$ "stuffed" hafnate pyrochlore is mainly due to the hydration of oxygen vacancies at 48f positions.

**Keywords:** SOFCs; hafnates; solid oxide electrolytes; ionic conductivity

## 1. Introduction

The (Ln = La-Lu; M = Ti, Zr, Hf) systems with a pyrochlore structure can be used as potential oxygen ion-conducting electrolytes for solid oxide fuel cell (SOFC) applications [1–16]. The most promising electrolytes among pyrochlores are rare-element (RE)-based zirconates: the Li-doped gadolinium zirconate Gd$_{1.7}$Li$_{0.3}$Zr$_2$O$_{6.7}$, with a conductivity of $1 \times 10^{-3}$ S/cm at 450 °C (~8 × 10$^{-3}$ S/cm at 700 °C) [1], and (Nd$_{2-x}$Zr$_x$)Zr$_2$O$_{7+x/2}$ ($x = 0.4$) ($1.78 \times 10^{-2}$ S/cm at 700 °C) [5]. It can be seen from previous results [4,5] that the solid solutions in broad isomorphism ranges with a variable Ln/M ratio (Ln = La-Lu; M = Ti, Zr, Hf) in the titanate, zirconate, and hafnate systems can be used to prepare materials with high oxygen ion conductivity. The undoubted advantage of some RE zirconate solid solutions with the pyrochlore structure is pure oxygen ion conductivity persisting in a wide temperature range (600–900 °C) and a wide range of oxygen partial pressures, and absence

of p-type conductivity at high oxygen partial pressure [4]. Undesirable electronic transport of materials used in PCFCs and PCECs leads to a decrease in their effectiveness [6,7].

The $Ln_2Hf_2O_7$ ($Ln$ = Sm, Eu, Gd) compounds have the pyrochlore structure, and the highest oxygen ion conductivity is reached for $Gd_2Hf_2O_7$: ~$1.3 \times 10^{-3}$ S/cm at 750 °C [8,12]. The $Ln_2Hf_2O_7$ ($Ln$ = Dy, Y, Ho, Yb) heavy RE hafnates have a fluorite structure, and the Dy- and Ho-based compounds have the highest conductivity among them [8]. These results were essentially confirmed by Sardar et al. [9], who were able to separate the total conductivity onto the bulk ($1.0 \times 10^{-4}$ S/cm at 700 °C) and grain boundary ($4.66 \times 10^{-5}$ S/cm at 700 °C) components for the $Ho_2Hf_2O_7$ fluorite prepared by the conventional solid-state reaction technique. Recently, Shlyakhtina et al. [16] synthesized $Eu_2(Hf_{2-x}Eu_x)O_{7-x/2}$ ($x = 0.1$) stuffed pyrochlores, which showed the highest oxygen ion conductivity among the hafnate family: ~$7.5 \times 10^{-4}$ S/cm at 700 °C. This is, however, lower than the conductivity of the RE zirconates and titanates [16–20]. The authors of [21] investigated the electronic structure of $La_2Hf_2O_7$ based on first-principle calculations with a particular focus on the anion Frenkel (anti-Frenkel) defect pair ($V_O^{\bullet\bullet} + O''_i$) in terms of defect structures with correlated formation energies. Such anion Frenkel pair ($V_O^{\bullet\bullet} + O''_i$) defects are the most stable ones in pure $La_2Hf_2O_7$. It is worth noting that, whereas the oxygen ion conductivity of the RE hafnates and related solid solutions has been the subject of intense research [8–16], there are few data on their proton conductivity, except for the $Nd_2(Hf_{2-x}Nd_x)O_{7-x/2}$ ($x = 0$; 0.1) system [22].

The proton conductivity of the $Ln_2Hf_2O_7$ hafnates and related solid solutions is essentially unexplored. Proton conduction was first found comparatively recently in $Nd_2(Hf_{2-x}Nd_x)O_{7-x/2}$ ($x = 0$; 0.1) neodymium hafnates [22]. Proton conductivity was shown both in neodimium hafnate ($1.25 \times 10^{-6}$ S/cm at 700 °C) and $Nd_2(Hf_{1.9}Nd_{0.1})O_{6.95}$ "stuffed" pyrochlore solid solution (~$1 \times 10^{-4}$ S/cm at 700 °C) [22]. Producing oxygen vacancies via acceptor substitutions is a well-known approach for raising the conductivity of various oxygen ion conductors [23–26]. The formation of vacancies in $La_2(Hf_{2-x}La_x)O_{7-x/2}$ ($x = 0.1$) as a result of substitution of the trivalent lanthanide on the tetravalent metal site can be represented as follows:

$$La_2O_3 \xrightarrow{HfO_2} 2La'_{Hf} + V_{O(48f)}^{\bullet\bullet} + 3O_O^x \tag{1}$$

The $Ln_2O_3$-$MO_2$ (M = Ti, Zr, Hf) systems have large isomorphism regions around the $Ln_2Ti_2O_7$ ($Ln$ = Sm-Lu), $Ln_2Zr_2O_7$ ($Ln$ = La-Eu), and $Ln_2Hf_2O_7$ ($Ln$ = La-Tb) pyrochlores. In a certain range of $Ln_2O_3$ concentrations, the pyrochlore solid solutions in these systems can transform into fluorite solid solutions (order–disorder transition) [27–32]. In the zirconate systems, these order–disorder ranges are symmetric regarding $Ln_2Zr_2O_7$, whereas in the titanate systems they are rather broad and are located mainly at high degrees of $Ln$ substitution for Ti at synthesis temperatures below 1700 °C. The investigation of the broad isomorphism ranges in the hafnate systems is complicated by the high synthesis temperatures of the solid solutions ($T_{syn.} \geq 1700$ °C) [22,32].

As shown in a precision XANES spectroscopy study [4], the broad isomorphism range in the $Tm_2(Ti_{2-x}Tm_x)O_{7-x/2}$ ($x = 0$–0.67) "stuffed" titanate system includes three different regions: (1) at $x = 0$–0.1, where only a P1 pyrochlore exists; (2) $x$~0.13–0.56, where P1 and P2 pyrochlore phases coexist, (P1—thulium-deficient; P2—thulium-enriched), and (3) $x$~0.56–0.67, where a defect fluorite F prevails. According to XRD data, in the $Yb_2(Ti_{2-x}Yb_x)O_{7-x/2}$ system, two pyrochlore phases coexist in a range of $x = 0.16$–0.37 [31].

In this work, ball milling was used to activate a starting oxide mixture for high-temperature synthesis of $La_2Hf_2O_7$ and $La_2(Hf_{1.9}La_{0.1})O_{6.95}$ solid solution with the aim of ascertaining whether they have proton conductivity. $La_2(Hf_{1.9}La_{0.1})O_{6.95}$ falls in the $La_2Hf_2O_7$–$La_2O_3$ isomorphous miscibility range, but a low degree of substitution ($x = 0.1$) simplifies the situation because in most of the $Ln_2M_2O_7$–$Ln_2O_3$ systems, only one solid solution exists at such degrees of substitution; in turn, this significantly facilitates interpretation of structural data and allows one to use diffraction techniques, without resorting to spectroscopy (Raman scattering or XANES) [2–4].

## 2. Materials and Methods

$La_2Hf_2O_7$ and $La_2(Hf_{1.9}La_{0.1})O_{6.95}$ were synthesized by reacting appropriate oxide mixtures ($La_2O_3 + HfO_2$) after mechanical activation in an Aronov ball mill [33,34]. The parameters of the Aronov ball mill were as follows: an oscillation amplitude of 0.5 cm, a frequency of 50 Hz, a vial volume of 120 $cm^3$, and a ball/powder weight ratio of 15. The $La_2O_3$ powder resource was annealed at 1000 °C for 2 h to remove absorber gases and then placed in a desiccator after cooling to 850 °C. The calcined mixtures were milled and pressed at 650 MPa and then sintered at 1600 °C for 10 h in air. The relative densities of the obtained ceramic samples were determined by measuring their weight and dimensions; they were found to be 92% above the theoretical levels for both $La_2Hf_2O_7$ and $La_2(Hf_{1.9}La_{0.1})O_{6.95}$ compounds. In the case of $La_2(Hf_{2-x}La_x)O_{7-x/2}$ ($x = 0.1$) "stuffed" hafnate pyrochlore, the ceramic's relative density was determined by hydrostatic weighing in toluene (94.8%). The phase composition features and structural properties of the powder samples were analyzed by the X-ray diffraction (XRD) technique using a DRON-3M diffractometer (Bragg-reflection geometry, Cu $K_\alpha$ radiation) in the range $2\theta = 10°-75°$ at ambient condition.

Scanning electron microscopy (SEM—using a JEOL JSM-6390LA device) was employed to analyze the morphology of the ceramic samples as well as to estimate their chemical compositions. Energy dispersive X-ray microanalysis (EDX) was used to obtain X-ray concentration maps of elements.

The $La_2(Hf_{2-x}La_x)O_{7-x/2}$ ($x = 0$, 0.1) ceramics were subject to thermogravimetric analysis in air using a NETZSCH STA 449C system (50–1000 °C, heating rate of 10 K/min, $Al_2O_3$ plate). To verify whether the $La_2(Hf_{1.9}La_{0.1})O_{6.95}$ ceramics contained structurally bound water, which might be responsible for proton conduction, we performed TG scans from 50 to 1000 °C in an oxygen atmosphere at a heating rate of 10 °C/min. To assess the reproducibility of the observed effects, the sample was reheated after cooling in the oxygen atmosphere.

The disk-shaped ceramic samples (diameter ~9 mm and thickness ~2 mm) were fabricated for electrical measurements. Electrode contacts to the sample were made by firing ChemPur C3605 paste with colloidal platinum at 950 °C for 0.5 h. The conductivity of $La_2Hf_2O_7$ and $La_2(Hf_{1.9}La_{0.1})O_{6.95}$ was studied by impedance spectroscopy analysis (EIS) via a two-probe AC method in both dry and wet air atmospheres. A combination of P-5X potentiostat/galvanostat and a frequency response analyzer module (Electrochemical Instruments Ltd., Chernogolovka, Russia) was purposefully used to provide electrical conductivity measurements. The impedance spectra between 100 and 900 °C were recorded over a frequency range of $10^{-1}$–$5 \times 10^6$ Hz at a signal amplitude of 150 mV. The dry atmosphere was created by passing air through a KOH, while the wet atmosphere was regulated by passing air through a water saturator held at 20 °C, which ensured a constant humidity of ~0.023 atm (2.3% $H_2O$). Air flow rate was 130 mL/min. To obtain stable water vapor pressure before measurement, the sample was kept at each temperature for 40 min. The impedance data fitting was performed using Zview software (Scribner Associates, Southern Pines, NC, USA).

In addition, the total conductivity of $La_2(Hf_{1.9}La_{0.1})O_{6.95}$ the was evaluated by the four-probe DC method. These measurements were performed in a temperature range of 500–900 °C in dry and in wet oxidizing and reducing atmospheres (air and $N_2$). Wet atmospheres ($P_{H2O} = 0.02$ atm) were generated by bubbling gases through water thermostatted at a temperature of 18 °C.

## 3. Results and Discussion

### 3.1. Study of the $La_2Hf_2O_7$ and $La_2(Hf_{1.9}La_{0.1})O_{6.95}$ Structure by the X-ray Diffraction Method

The pyrochlore structure of $La_2Hf_2O_7$ has two distinct cation sites—those of La, in the center of a distorted $LaO_8$ cube (scalenohedron), and Hf, in the center of a distorted $HfO_6$ octahedron (trigonal antiprism)—and three distinct anion sites: 48f (O1), 8a (O2), and 8b (O3). Each O1 oxygen (48f) is surrounded by two $La^{3+}$ and two $Hf^{4+}$ cations, each O2 oxygen (8a) is coordinated by four $La^{3+}$ cations, and each O3 oxygen, which is always

empty in ordered stoichiometric pyrochlores, is coordinated by four $Hf^{4+}$ cations. $La_2Hf_2O_7$ is the most ordered pyrochlore phase in the $Ln_2Hf_2O_7$ ($Ln$ = La-Tb) series, without cation disordering. In the case of $La_2(Hf_{1.9}La_{0.1})O_{6.95}$, the Hf-to-La substitution leads to a decrease in the intensity of the 111, 311, 331, and 531 pyrochlore superstructure reflections compared with $La_2Hf_2O_7$ (Figure 1a, scans 1 and 2, respectively). In addition, the cubic pyrochlore cell parameter increases from 10.7731(2) Å for nominally stoichiometric $La_2Hf_2O_7$ to 10.7833 (1) Å for $La_2(Hf_{1.9}La_{0.1})O_{6.95}$ because the ionic radius of lanthanum in the center position of the distorted octahedra exceeds that of hafnium (R $La^{3+}_{CN6}$= 1.032Å; R $Hf^{4+}_{CN6}$ = 0.71Å).

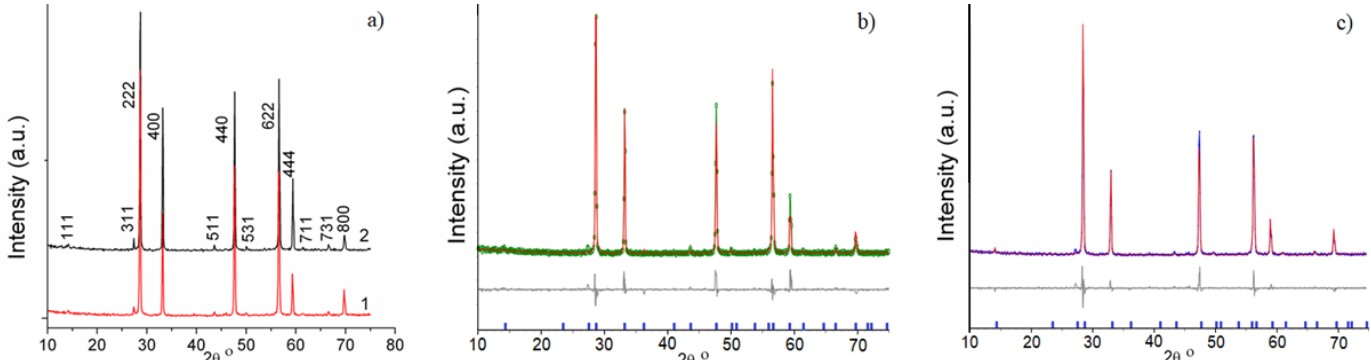

**Figure 1.** (**a**) X-ray diffraction patterns of (1) $La_2Hf_2O_7$ and (2) $La_2(Hf_{1.9}La_{0.1})O_{6.95}$; (**b**) Rietveld refinement of $La_2Hf_2O_7$ X-ray diffraction pattern: the experimental data (green dots), the calculated data (red line), and the difference between the experimental and calculated data (gray line); (**c**) Rietveld refinement of $La_2(Hf_{1.9}La_{0.1})O_{6.95}$ X-ray diffraction pattern: the experimental data (blue line), the calculated data (red line), and the difference between the experimental and calculated data (gray line).

A detailed analysis of the phase composition and crystal structure of the obtained materials was carried out by refining diffraction patterns by the Rietveld method (by means of the TOPAS software). The results of such a refinement are summarized in Table 1 and in Figure 1b,c.

**Table 1.** Rietveld refinement of positions for $La_2(Hf_{2-x}La_x)O_{7-x/2}$ ($x$ = 0, 0.1) (sp.gr. $Fd$-$3m$) at T = 300 K.

| c | Site | $x$ | $y$ | $z$ | Occ. | $R_{exp}$, % $R_{wp}$, % $R_p$, % GOF | Unit Cell Parameter $a$, Å | Crystal Density, g/cm³ |
|---|---|---|---|---|---|---|---|---|
| $La_2Hf_2O_7$ | $La_{La}$(16d) | 0.5000 | 0.5000 | 0.5000 | 1 | 9.61 | | |
| | $Hf_{Hf}$(16c) | 0.0000 | 0.0000 | 0.0000 | 1 | 12.42 | | |
| | $La_{Hf}$(16c) | 0.0000 | 0.0000 | 0.0000 | 0 | 9.93 | 10.7731(2) | 7.93 |
| | O(1)(48f) | 0.3586 | 0.1250 | 0.1250 | 1 | 1.26 | | |
| | O(2)(8a) | 0.1250 | 0.1250 | 0.1250 | 1 | | | |
| $La_2(Hf_{1.9}La_{0.1})O_{6.95}$ | $La_{La}$(16d) | 0.5000 | 0.5000 | 0.5000 | 1 | 9.45 | | |
| | $Hf_{Hf}$(16c) | 0.0000 | 0.0000 | 0.0000 | 0.95 | 15.08 | | |
| | $La_{Hf}$(16c) | 0.0000 | 0.0000 | 0.0000 | 0.05 | 11.88 | 10.7833(1) | 7.86 |
| | O(1)(48f) | 0.3770 | 0.1250 | 0.1250 | 0.992 | 1.6 | | |
| | O(2)(8a) | 0.1250 | 0.1250 | 0.1250 | 1 | | | |

According to the XRD data, the $La_2(Hf_{1.9}La_{0.1})O_{6.95}$ sample is a single-phase pyrochlore with no detectable secondary phase. This result agrees well with Durand's data [35], according to which the homogeneity region of this solid solution extends to $x$ = 0.16 at a temperature of 1600 °C.

Similar results were previously obtained for $Nd_2(Hf_{2-x}Nd_x)O_{7-x/2}$ ($x$ = 0, 0.1) synthesized at 1600 °C [22]. The incorporation of 5% La into the Hf sublattice leads to the

formation of oxygen vacancies and a defect pyrochlore structure according to the following scheme:

$$La_2O_3 + Hf^x_{Hf} + 1/2O^x_O \rightarrow La'_{Hf} + 1/2V^{\bullet\bullet}_O + 1/2La_2Hf_2O_7 \quad (2)$$

According to Mullens et al. [2], only one pyrochlore phase exists in the $Tm_2(Ti_{2-x}Tm_x)O_{7-x/2}$ ($x = 0$–0.67) system at a low substitution degree, $x = 0$–0.1, unlike at higher degrees of substitution ($x > 0.1$) [2]. As shown for $A_2(B_{2-x}A_x)O_{7-x/2}$ "stuffed" pyrochlores [2] using the $Tm_2(Ti_{2-x}Tm_x)O_{7-x/2}$ ($x = 0$–0.67) system as an example, at low degrees of Ti-for-Tm substitution (at $x = 0$–0.1), oxygen vacancies are formed predominantly in position 48f, opposed to the change in the vacancies formation in two positions (48f and 8b) in $Y_2(Zr_yTi_{1-y})_2O_7$ [36] and $Y_2Ti_{2-x}Hf_xO_7$ [37] as a result of isovalent substitutions in the tetravalent metal sublattice. Note that in the case of the formation of anti-site pairs in nominally stoichiometric pyrochlores at high temperatures (e.g., in the oxygen- and proton-conducting neodymium hafnate $Nd_2Hf_2O_7$ synthesized at 1700 °C), oxygen vacancies are formed in position 48f [22].

It can be assumed that there a similar situation obtains for the $Ln_2(Hf_{2-x}Ln_x)O_{7-x/2}$ ($Ln$ = La, Nd; $x = 0.1$) hafnate systems and the $Tm_2(Ti_{2-x}Tm_x)O_{7-x/2}$ ($x = 0$–0.1) titanate system. $La_2(Hf_{1.9}La_{0.1})O_{6.95}$ "stuffed" pyrochlore was shown to contain oxygen vacancies only in position 48f (Table 1).

### 3.2. Microstructure of $La_2Hf_2O_7$ and $La_2(Hf_{1.9}La_{0.1})O_{6.95}$

Table 2 presents the La/Hf ratio on the outer and fracture surfaces of $La_2Hf_2O_7$ and $La_2(Hf_{1.9}La_{0.1})O_{6.95}$. The La/Hf ratio on the fracture surfaces is similar to the intended one, whereas the outer surface is found to be enriched with lanthanum. Note that the nominally stoichiometric $La_2Hf_2O_7$ ceramic is visually inhomogeneous and two-colored; its microstructure is presented in Figure 2a,b. Its surface is white, which correlates with the $La_2O_3$ excess detected by EDS/EDX (energy dispersive X-ray spectroscopy X-ray analysis), and there is a brown layer in its interior. The micrograph of a fracture surface of the ceramic in Figure 2a demonstrates its inhomogeneity. Nevertheless, the $La_2Hf_2O_7$ ceramic is dense: its relative density is 7.4 g/cm³, i.e., 93.3% of its theoretical density. The $La_2(Hf_{1.9}La_{0.1})O_{6.95}$ ceramic is light brown in colour and more homogeneous than the $La_2Hf_2O_7$ ceramic prepared under the same conditions. The La/Hf ratio is near the expected one (Table 2). The microstructure of the $La_2(Hf_{1.9}La_{0.1})O_{6.95}$ ceramic is shown in Figure 2c,d. Note the La excess is observed on the outer surface of the $La_2Hf_2O_7$ compared to $La_2(Hf_{1.9}La_{0.1})O_{6.95}$ "stuffed" pyrochlore (Table 2). The density of the $La_2(Hf_{1.9}La_{0.1})O_{6.95}$ ceramic is 7.27 g/cm³, which is 92.5% of its theoretical density, whereas its density determined by hydrostatic weighing is higher: 94.8%.

**Table 2.** Average La/Hf ratio on the outer and fracture surfaces of ceramics: SEM/EDX ten point analysis of $La_2(Hf_{2-x}La_x)O_{7-x/2}$ ($x = 0$, 0.1).

| Composition | Analysing Area | Nominal Stoichiometry La/Hf | La/Hf Mean | Standard Error of the Mean |
|---|---|---|---|---|
| $La_2(Hf_{1.9}La_{0.1})O_{6.95}$ | Outer surface | 1.11 | 1.34 | 0.04 |
| | Fracture surface | | 1.24 | 0.05 |
| $La_2Hf_2O_7$ | Outer surface | 1.00 | 1.57 | 0.13 |
| | Fracture surface | | 1.05 | 0.04 |

The average La/Hf ratio value on the fracture surface exceeds the nominal stoichiometry within the standard error of the mean (SEM), but it is close to it and significantly less than this ratio on the outer surface (Table 2). In addition, visual information about the inhomogeneity of the color of the samples cannot serve as a basis for considering the samples as polyphasic; however, based on the EDX analysis data, we can conclude that

there is a gradient in the La/Hf ratio. This inhomogeneity is apparently associated with significant inertia of the system, as a result of which the diffusion of atoms proceeds slowly.

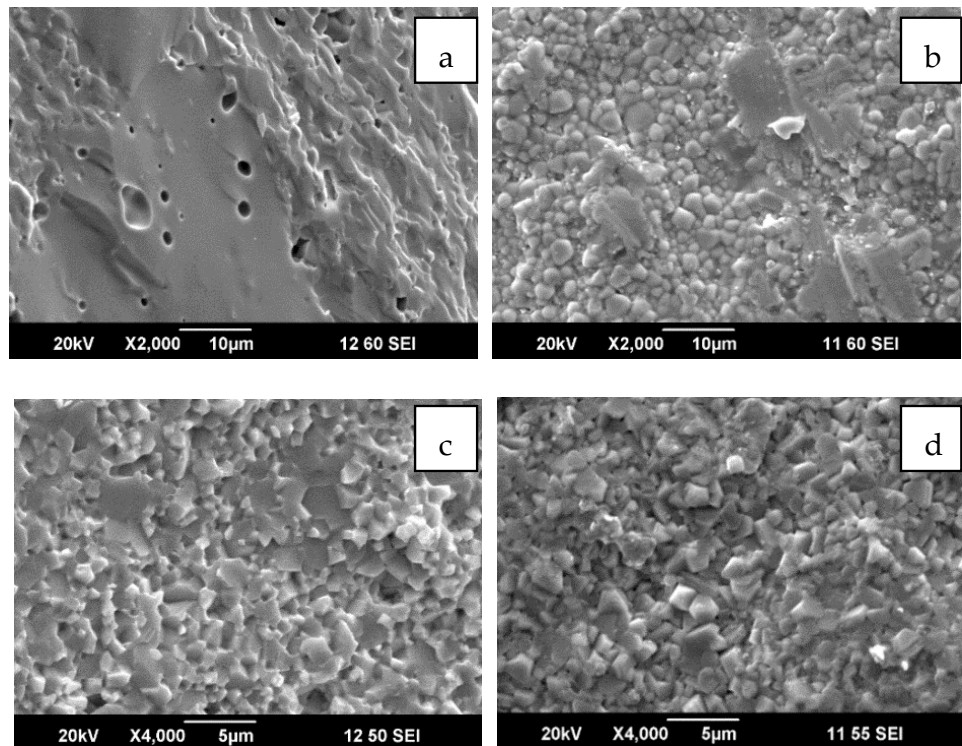

**Figure 2.** Microstructure of (**a,c**) a fracture surface and (**b,d**) the outer surface of the (**a,b**) $La_2Hf_2O_7$ and (**c,d**) $La_2(Hf_{1.9}La_{0.1})O_{6.95}$ ceramics.

### 3.3. Oxygen Ion and Proton Conductivity of $La_2(Hf_{2-x}La_x)O_{7-x/2}$ (x = 0; 0.1) in Dry and Wet Air

Electrochemical impedance spectroscopy data for the $La_2Hf_2O_7$ ceramic demonstrate that its impedance response corresponds to dielectric behaviour. No significant conductivity was detected in the range 300–900 °C with our measurement equipment. Therefore, $La_2Hf_2O_7$ is a dielectric, unlike $Nd_2Hf_2O_7$, studied previously [22], whose conductivity in wet air is ~1.25 × 10$^{-6}$ S/cm at 700 °C, approaching the proton conductivity of $La_2Zr_2O_7$ [38]. The negligible conductivity of $La_2Hf_2O_7$ is attributable to the high covalence of the La–Hf bonds and the perfect structure of this pyrochlore phase. Figure 3 shows Arrhenius plots for a number of $Ln_2(Hf_{1.9}Ln_{0.1})O_{6.95}$ ((1) $Ln$ = La; (2) Nd [22]; (3) Sm [16]; (4) Eu) [16] "stuffed" pyrochlores in dry and wet air. $La_2(Hf_{1.9}La_{0.1})O_{6.95}$ has considerable proton conductivity in the range 400–750 °C. The impedance spectra of $La_2(Hf_{1.9}La_{0.1})O_{6.95}$ at 530 and 700 °C in dry and wet air are presented in Figure S1. Apparent activation energy ($E_a$) of total conductivity temperature dependences of $Ln_2(Hf_{1.9}Ln_{0.1})O_{6.95}$ (Ln = La, Nd, Sm, Eu) "stuffed" hafnate pyrochlores in dry and wet air are presented in Table 3. The conductivity of $La_2(Hf_{1.9}La_{0.1})O_{6.95}$ in wet air is ~8 × 10$^{-5}$ S/cm at 700 °C (Figure 3), which differs very little from the conductivity of $Nd_2(Hf_{1.9}Nd_{0.1})O_{6.95}$ under the same conditions [22]. The conductivity of $Ln_2(Hf_{1.9}Ln_{0.1})O_{6.95}$ (Ln = Sm, Eu) is not higher in wet air [16], which indicates that Sm- and Eu-based "stuffed" hafnate pyrochlores are purely oxygen ion conductors (Figure 3). The $Ln_2(Hf_{1.9}Ln_{0.1})O_{6.95}$ (Ln = La, Nd) "stuffed" pyrochlores, in which the major defects are oxygen vacancies in position 48f, have proton conductivity due to interaction of the oxygen vacancies in position 48f with water according to the Equation (3):

$$H_2O + V_O^{\bullet\bullet} + O_O^x \rightarrow 2OH_O^{\bullet} \tag{3}$$

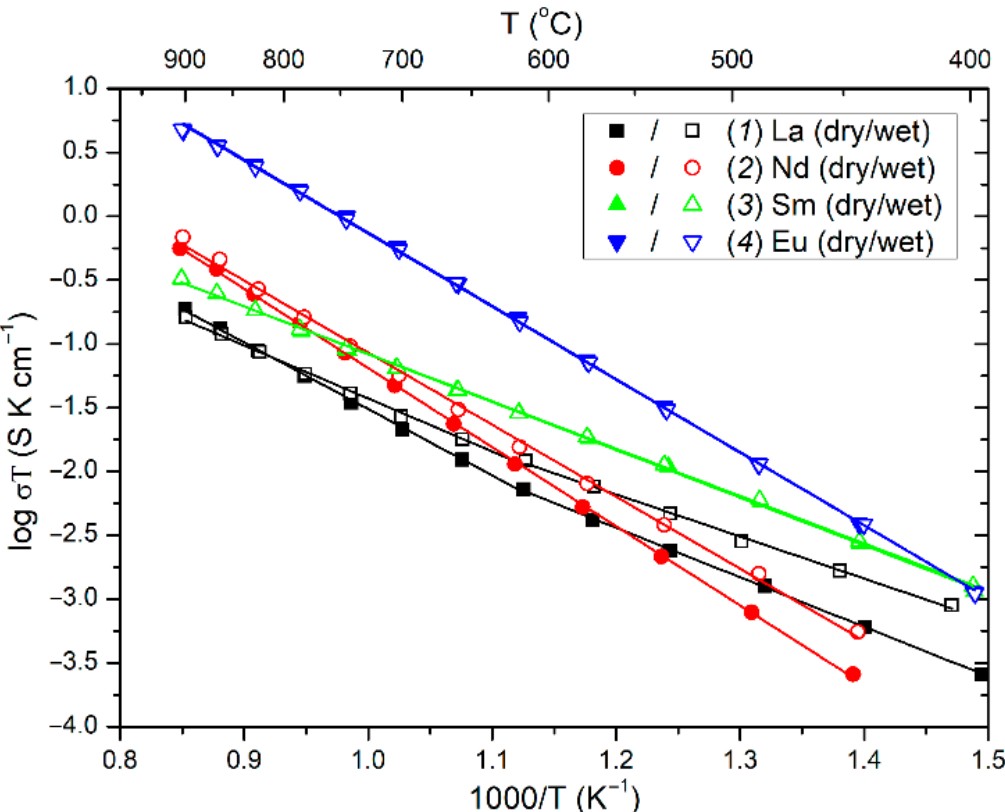

**Figure 3.** Arrhenius plots of conductivity for different $Ln_2(Hf_{1.9}Ln_{0.1})O_{6.95}$ "stuffed" hafnate pyrochlores ((1) $Ln$ = La; (2) Nd [22]; (3) Sm [16]; (4) Eu) [16] in dry and wet air.

**Table 3.** Apparent activation energy ($E_a$) of total conductivity temperature dependences of $Ln_2(Hf_{1.9}Ln_{0.1})O_{6.95}$ (Ln = La, Nd, Sm, Eu) stuffed hafnate pyrochlores in dry and wet air.

| Composition $Ln_2(Hf_{1.9}Ln_{0.1})O_{6.95}$ | Atmosphere | Temperature Range, °C | $E_a$ ($\pm$0.01), eV | References |
|---|---|---|---|---|
| La | Dry air | 300–600 | 0.77 | This work |
| | | 600–900 | 1.04 | |
| | Wet air | 300–600 | 0.66 | |
| | | 600–900 | 0.82 | |
| Nd | Dry air | 550–900 | 1.25 | [22] |
| | Wet air | 550–900 | 1.16 | |
| Sm | Dry air | 400–900 | 0.74 | [16] |
| | Wet air | 400–900 | 0.74 | |
| Eu | Dry air | 400–900 | 1.14 | [16] |
| | Wet air | 400–900 | 1.14 | |

It is obvious (Table 3) that below 800 °C the activation energies for conduction in the $Ln_2(Hf_{1.9}Ln_{0.1})O_{6.95}$ (Ln = La, Nd) "stuffed" hafnate pyrochlores in wet air are lower than those in dry air. This is usual proton conductor behaviour. $Ln_2(Hf_{1.9}Ln_{0.1})O_{6.95}$ (Ln = Sm, Eu) "stuffed" pyrochlores have the same activation energy (Table 3) for conductivity, which is independent of humidity.

*3.4. DSC and TG Characterization of La₂(Hf₁.₉La₀.₁)O₆.₉₅ "Stuffed" Pyrochlore*

　　No significant proton conductivity for the $Ln_2(Hf_{1.9}Ln_{0.1})O_{6.95}$ ($Ln$ = Sm, Eu) materials is attributed to the low basicity of $Sm_2O_3$ and $Eu_2O_3$ and the low hydration degree of oxygen vacancies in position 48f in their pyrochlore solid solutions compared to La- and Nd-counterparts [39]. The $La_2(Hf_{1.9}La_{0.1})O_{6.95}$ "stuffed" pyrochlore powder was heated twice up to 1000 °C. Figure 4a shows DSC and TG curves of this sample during the first (curves 1 and 1a) and second (curves 2 and 2a) heating cycles. The total weight loss between 50 and 1000 °C is 0.41% in the first and 0.32% in the second heating cycles. The TG curve obtained during the first heating has two steps. A sharpest weight loss is observed between 50 and 650 °C, with $\Delta M_1$ = 0.33%. According to Colomban [40], this temperature range corresponds to the removal of surface water and hydroxyl ions. Above 650 °C, water was removed rather slowly, up to 1000 °C. This weight loss is related to the structurally bounded water and interstitial protons [40]. The weight loss was about $\Delta M_1$ = 0.08% at 650–1000 °C. After cooling in the same oxygen atmosphere, the sample was reheated, and the weight loss between 50 and 650 °C was 0.27%, i.e., smaller than in the first heating cycle. The slope of the second weight loss step, between 650 and 1000 °C, also decreased, and the weight loss was $\Delta M_1$ = 0.05%, i.e., also smaller than in the first heating cycle. Thus, the second heating showed that, during cooling in an oxygen atmosphere, the material regained most of the physically and structurally bound water. The present results demonstrate that $La_2(Hf_{1.9}La_{0.1})O_{6.95}$ "stuffed" pyrochlore powder rehydrates rather easily during cooling and the water is then present in the form of not only physically bound water but structurally bound water and interstitial protons.

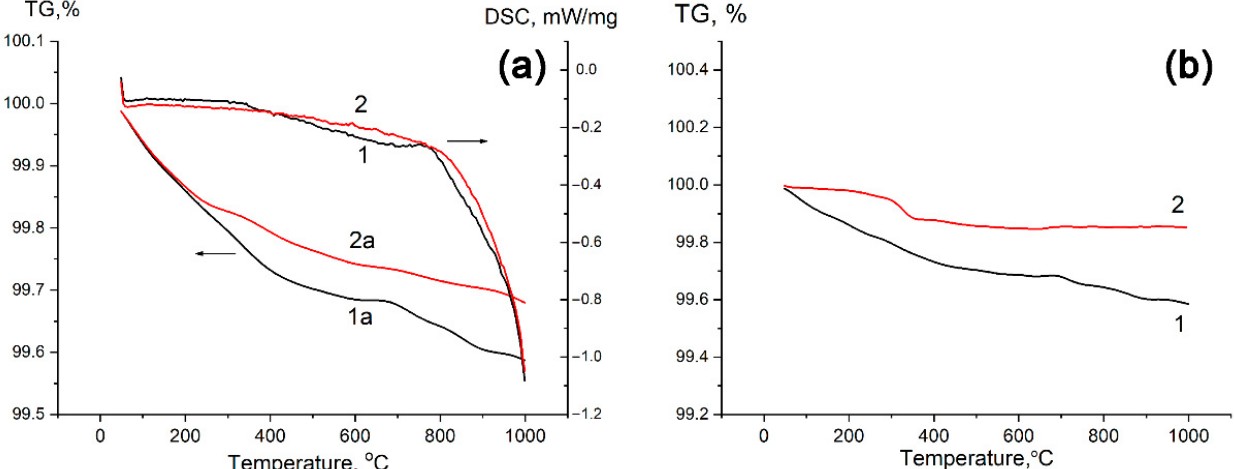

**Figure 4.** (**a**) DSC curves (1—first heating; 2—second heating) and TG curves (1a—first heating; 2a—second heating) of the powder sample $La_2(Hf_{1.9}La_{0.1})O_{6.95}$ (left arrow—TG axis, right arrow—DSC axis); (**b**) TG curves: (1) first heating of $La_2(Hf_{1.9}La_{0.1})O_{6.95}$ and (2) $La_2Hf_2O_7$ powder sample.

　　Figure 4b displays TG curves of the $La_2(Hf_{1.9}La_{0.1})O_{6.95}$ "stuffed" pyrochlore powder (curve 1) and pure $La_2Hf_2O_7$ powder (curve 2). It can easily be seen that $La_2Hf_2O_7$ loses surface water and hydroxyl ions up to 650 °C and above this temperature there is no weight loss, while $La_2(Hf_{1.9}La_{0.1})O_{6.95}$ loses structurally bound water and protons in the 650–1000 °C temperature interval ($\Delta M_1$ = 0.08%). In this work, the sharpest weight loss (50–650 °C) for $La_2Hf_2O_7$ (Figure 4b, curve 2) is associated with the release of $H_2O$ and $CO_2$. This can be partially assigned to the decomposition of La-containing carbonates and hydroxycarbonates formed due to $CO_2$ trapping from air during cooling and located at grain boundaries or in pores. The exoeffect on the DSC curve of $La_2(Hf_{1.9}La_{0.1})O_{6.95}$ near 800 °C (Figure 4a, curve 1) can be caused by the structure relaxation after the removal of

bulk $CO_3{}^{2-}$ groups. It should be noted that after the second heating this effect disappeared (Figure 4a, curve 2).

### 3.5. Four-Probe Proton Conductivity Measurements for the La$_2$(Hf$_{1.9}$La$_{0.1}$)O$_{6.95}$ Stuffed Pyrochlore in Dry and Wet Nitrogen and Air

The four-probe conductivity measurement results for La$_2$(Hf$_{1.9}$La$_{0.1}$)O$_{6.95}$ are presented in Figure 5. In the measurements, we used a bar-shaped sample cut from the center of the ceramic pellet. It can be seen that the conductivity in both wet air and wet N$_2$ exceeds that in the dry atmospheres. The conductivity of La$_2$(Hf$_{1.9}$La$_{0.1}$)O$_{6.95}$ in wet air is $\sim 8 \times 10^{-5}$ S/cm at 700 °C, in agreement with the two-probe measurement results (Figure 3). The total conductivity in wet nitrogen exceeds that in dry nitrogen, indicating that the La$_2$(Hf$_{1.9}$La$_{0.1}$)O$_{6.95}$ "stuffed" pyrochlore has proton conductivity. Clearly, the oxygen vacancies in position 48f of the La$_2$(Hf$_{1.9}$La$_{0.1}$)O$_{6.95}$ "stuffed" pyrochlore interact with water. A crucial role is played by the ability of the oxygen vacancies in position 48f, i.e., in the center of the $Ln^{3+}{}_2$Hf$^{4+}{}_2$ ($Ln$ = La, Nd) tetrahedra in the pyrochlore structure, to be hydrated, which is due to the basicity of lanthanum and neodymium compounds. The lower basicity of the $Ln_2$(Hf$_{1.9}$Ln$_{0.1}$)O$_{7-x/2}$ ($Ln$ = Sm, Eu) "stuffed" pyrochlores makes them purely oxygen ion conductors [16].

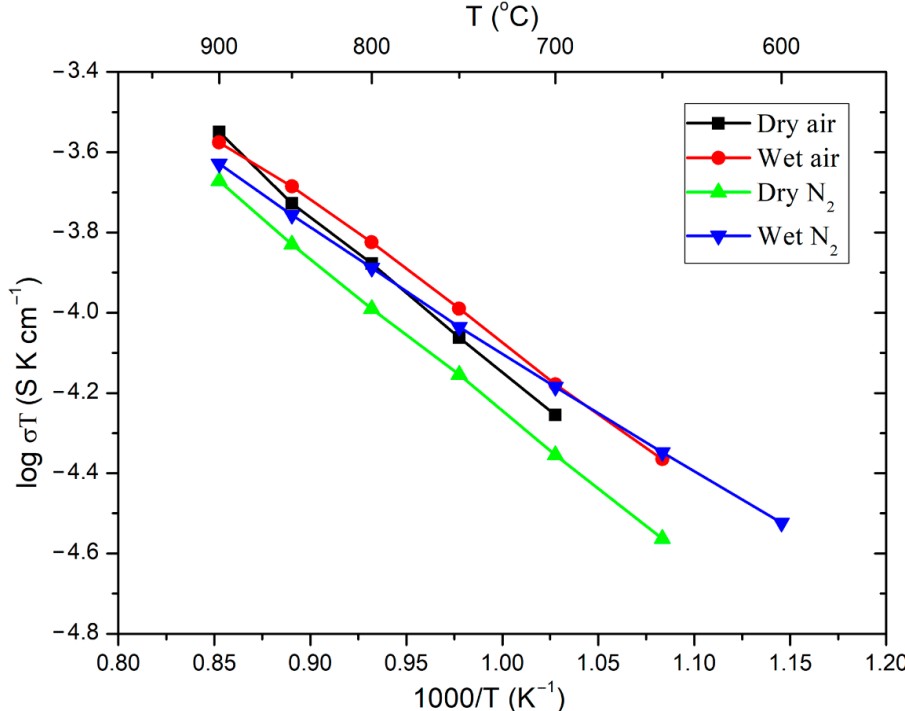

**Figure 5.** Total conductivity of La$_2$(Hf$_{1.9}$La$_{0.1}$)O$_{6.95}$ in dry and wet air and N$_2$.

La$_2$(Hf$_{1.9}$La$_{0.1}$)O$_{6.95}$ "stuffed" hafnate pyrochlores demonstrated proton contribution $\sim 1 \times 10^{-5}$ S/cm at 600 °C, whereas the conductivity of Ca-doped zirconate pyrochlores is higher by $\sim 1$–1.5 orders of magnitude. For example, the conductivity of (La$_{2-x}$Ca$_x$)Zr$_2$O$_{7-\delta}$ ($x = 0.05$) in wet air is $7 \times 10^{-4}$ S/cm at 600 °C while that of (Nd$_{2-x}$Ca$_x$)Zr$_2$O$_{7-\delta}$ ($x = 0.05$) is $2.5 \times 10^{-4}$ S/cm at 600 °C) and that of (Sm$_{2-x}$Ca$_x$)Zr$_2$O$_{7-\delta}$ ($x = 0.05$) is $7.5 \times 10^{-4}$ S/cm at 600 °C [41–45]. The comparison of the proton conductivity of the Ca-doped zirconates and "stuffed" hafnate pyrochlores indicates that the higher basicity of cations around an oxygen vacancy in position 48f as a result of Ca-substitution, a more basic cation, for the lanthanide increases the hydration at this position and, as a result, proton conductivity rises by almost two orders of magnitude [41–48]. Another factor contributing to the decrease in the conductivity of the "stuffed" hafnate pyrochlores is the presence of more covalent and strong Hf-O bonds in hafnates relative to zirconates [32].

## 4. Conclusions

$La_2Hf_2O_7$ and a $La_2(Hf_{1.9}La_{0.1})O_{6.95}$ solid solution were prepared using ball milling of oxide mixtures followed by high-temperature annealing at 1600 °C for 10 h in air. The prepared $La_2Hf_2O_7$ ceramic is nonuniform in composition, with an excess of lanthanum oxide on its surface. The $La_2Hf_2O_7$ ceramic is a dielectric, whereas the $La_2(Hf_{1.9}La_{0.1})O_{6.95}$ ceramic is more homogeneous and has a defect pyrochlore structure type with oxygen vacancies at the 48f positions of the pyrochlore structure. $La_2(Hf_{1.9}La_{0.1})O_{6.95}$ "stuffed" pyrochlore exhibits both oxygen ion and proton transference. The use of impedance spectroscopy in different atmospheres (dry and wet air and nitrogen) with two-probe AC and four-probe DC measurements provided reliable data on the proton conductivity of the $La_2(Hf_{1.9}La_{0.1})O_{6.95}$ "stuffed" hafnate pyrochlore. The $La_2(Hf_{1.9}La_{0.1})O_{6.95}$ "stuffed" pyrochlore has been shown to have both oxygen-ion and proton conductivity in the range 600–900 °C, with a proton conductivity of $\sim 8 \times 10^{-5}$ S/cm at 700 °C. The conductivity in the wet $N_2$ atmosphere was higher than in the dry $N_2$ atmosphere in the temperature range of 600–900 °C, which indicates hydration of oxygen vacancies at 48f positions. In the air atmosphere, the difference was smaller, since some of the oxygen vacancies serve to move oxygen ions.

Clearly, the excess conductivity in a wet atmosphere relative to a dry one for the $La_2(Hf_{1.9}La_{0.1})O_{6.95}$ "stuffed" hafnate pyrochlore is due solely to proton conductivity, without any contribution from surface conduction.

**Supplementary Materials:** The following supporting information can be downloaded at: https://www.mdpi.com/article/10.3390/app12094342/s1, Figure S1: The impedance spectra of $La_2(Hf_{1.9}La_{0.1})O_{6.95}$ at 530 and 700 °C in dry and wet air.

**Author Contributions:** A.V.S.: conceptualization, methodology, writing—original draft preparation, review and editing; N.V.L., G.A.V. and G.E.N.: investigation, formal analysis; A.V.K. and D.N.S.: investigation, data curation, visualization; I.V.K.: resources; D.A.M.: formal analysis, writing—review and editing. All authors have read and agreed to the published version of the manuscript.

**Funding:** This research received no external funding.

**Institutional Review Board Statement:** Not applicable.

**Informed Consent Statement:** Not applicable.

**Data Availability Statement:** Not applicable.

**Acknowledgments:** The support of this work by the Russian Science Foundation (Project 18-13-00025) is gratefully acknowledged. The 50% conductivity measurements of samples were performed in accordance with the state task for IPCP RAS, state registration No. AAAA-A19-119061890019-5.

**Conflicts of Interest:** The authors declare no conflict of interest.

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
