# Peer review of "Proton Conductivity of La2(Hf2−xLax)O7−x/2 “Stuffed” Pyrochlores"

_applsci, doi:10.3390/app12094342_

Round 1
Reviewer 1 Report
This work reports on "Proton conductivity of La2 (Hf2-x Lax )O 7-x/2 “stuffed” pyrochlores". The authors demonstrate that La2(Hf2-xLax)O7-x/2 (x = 0.1) “stuffed” pyrochlore has both oxygen-ion and proton conductivities in the temperature range of 400–900 °C. The proton conductivity level is found to be equal to ~ 8×10- 5 S/cm at 700 °C. Based on the conductivity test results in a wet/dry atmosphere, the authors explain the origin of its proton conductivity is mainly due to hydration of oxygen vacancies in 48f positions. The motivation of the work, including the differentiation with the previous studies, was well introduced in this manuscript. Therefore, I recommend this paper to be published in Applied Sciences after making minor revisions. Some of the comments are shown below:
1. I suggest the authors denote a,b,c in Figure 1 and a,b in Figure 4.
2. Also, some figures (e.g., Figs. 1b-c) should be re-draw to be readable in the printed manuscript. The sizes of the data points and the fonts are too small.
3. Do you have references that can support your claim as follow?
- Line 311: "Another factor contributing to the decrease in the conductivity of the “stuffed” hafnate pyrochlores is more covalent and strange Hf-O bond in hafnates relative to zirconates."
Author Response
Rev. 1. Comments and Suggestions for Authors
This work reports on "Proton conductivity of La2 (Hf2-x Lax )O 7-x/2 “stuffed” pyrochlores". The authors demonstrate that La2(Hf2-xLax)O7-x/2 (x = 0.1) “stuffed” pyrochlore has both oxygen-ion and proton conductivities in the temperature range of 400–900 °C. The proton conductivity level is found to be equal to ~ 8×10- 5 S/cm at 700 °C. Based on the conductivity test results in a wet/dry atmosphere, the authors explain the origin of its proton conductivity is mainly due to hydration of oxygen vacancies in 48f positions. The motivation of the work, including the differentiation with the previous studies, was well introduced in this manuscript. Therefore, I recommend this paper to be published in Applied Sciences after making minor revisions. Some of the comments are shown below:
- I suggest the authors denote a,b,c in Figure 1 and a,b in Figure 4.
Reply: Thank you! We have changed Figs.1 and 4 in accordance with your suggestions.
- 2. Also, some figures (e.g., Figs. 1b-c) should be re-draw to be readable in the printed manuscript. The sizes of the data points and the fonts are too small.
Reply: Done. The figures have been re-draw following your suggestions.
- Do you have references that can support your claim as follow?
- Line 311: "Another factor contributing to the decrease in the conductivity of the “stuffed” hafnate pyrochlores is more covalent and strange Hf-O bond in hafnates relative to zirconates."
Reply: Thank you for your comment. There was a typo in this sentence.
Correct sentence is "Another factor contributing to the decrease in the conductivity of the “stuffed” hafnate pyrochlores is more covalent and strong Hf-O bond in hafnates relative to zirconates [32]".

Reviewer 2 Report
The manuscript entitled “Proton conductivity of La2(Hf2-xLax)O7-x/2 “stuffed” pyrochlores” by A.V. Shlyakhtina et al. is quite interesting, but before publishing requires several corrections. In the current state the conclusions are not reliably justified.
- The first unacceptable thing which may be easily noticed is too high number of the cited references co-authored by the present manuscript authors (17 out of 45). It should be noted that the google scholar search “stuffed pyrochlores” yields the first mention of the name Shlyakhtina in the 53rd The other feature regarding references may be illustrated by the sentence “It can be seen from previous results [4, 5] that the solid solutions in broad iso- morphism ranges with a variable Ln/M ratio (Ln = La-Lu; M = Ti, Zr, Hf) in the titanate, zirconate, and hafnate systems can be used to prepare materials with high oxygen ion conductivity [2-16].” Giving the references as [2-16] is completely useless for a reader. I would recommend giving particular number in its appropriate position (related to particular composition).
- Why the XRD patterns, especially those for the Rietveld refinement were taken only up to 75 degrees?
- Knowing that the La2H2O7 samples were composed of (as interpreted by the authors) the surface La2O3 layer and apparently La-depleted inside pyrochlore, what meaning may be attributed to the conductivity measurements of that sample? Was I measured after removing the top layer?
- Did the authors check the surface of La2H2O7 samples with XRD (before grinding)?
- Is the sentence “Note the slight excess of lanthanum on the surface of the La2(Hf2-xLax)O7-x/2 (x = 0.1) stuffed pyrochlore compared to La2Hf2O7 (Table 2)” correct? Table 2 shows more lanthanum on the surface of La2Hf2O7.
- The authors show TG results and state that “Thus, the second heating showed that, during cooling in an oxygen atmosphere, the material regained most of the physically and structurally bound water.” and “It is well seen that La2Hf2O7 loses surface water and hydroxyl ions up to 650 ºC and above this temperature there is no weight loss, while La2(Hf2-xLax)O7-x/2 (x = 0.1) loses structurally bound water 283 and protons in the 650 - 1000 °C temperature interval”. The processes of hydration/hydrogenation are usually studied in the atmosphere containing water vapour and/or during the change of atmosphere humidity. Simple loss/increase of the mass on heating/cooling does not prove that it is the “structurally bound water” and the concentration of proton defects that changes.
- The authors compare the values of the conductivity obtained with IS and the 4-probe DC method and conclude “ The 700 °C conductivity of La2(Hf2-xLax)O7-x/2 (x = 0.1) in wet air is ~ 8×10-5 S/cm, in agreement with the two-probe measurement results (Fig. 3).”. It seems to me, there is a mistake. Figure 5 shows that log(sigma x T) at 973 K (1.03 ) at wet atmosphere is approximately -4.17, whereas from Fig. 3 for the same temperature is approximately -1.7. It means about two orders of magnitude difference. Needless to say, I do not understand, why they should be similar. Why the authors do not compare activation energies?
Author Response
Rev.2 Comments and Suggestions for Authors
The manuscript entitled “Proton conductivity of La2(Hf2-xLax)O7-x/2 “stuffed” pyrochlores” by A.V. Shlyakhtina et al. is quite interesting, but before publishing requires several corrections. In the current state the conclusions are not reliably justified.
- The first unacceptable thing which may be easily noticed is too high number of the cited references co-authored by the present manuscript authors (17 out of 45). It should be noted that the google scholar search “stuffed pyrochlores” yields the first mention of the name Shlyakhtina in the 53rdThe other feature regarding references may be illustrated by the sentence “It can be seen from previous results [4, 5] that the solid solutions in broad iso- morphism ranges with a variable Ln/M ratio (Ln = La-Lu; M = Ti, Zr, Hf) in the titanate, zirconate, and hafnate systems can be used to prepare materials with high oxygen ion conductivity [2-16].” Giving the references as [2-16] is completely useless for a reader. I would recommend giving particular number in its appropriate position (related to particular composition).
Reply:
We have changed the citation.
We have reduced self-citations and included new references of another authors.
- Why the XRD patterns, especially those for the Rietveld refinement were taken only up to 75 degrees?
Reply: We used the Rietveld refinement solely to compare two samples. Therefore, for our study, it was important to process the diffraction patterns obtained under the same conditions. The error in determining the unit cell parameters and the x-parameter of 48f-oxygen for a larger range of angles chosen for the Rietveld refinement would not exceed 1%.
- Knowing that the La2H2O7 samples were composed of (as interpreted by the authors) the surface La2O3 layer and apparently La-depleted inside pyrochlore, what meaning may be attributed to the conductivity measurements of that sample? Was I measured after removing the top layer?
Reply: Thank you for your comment. The conductivity of samples was measured by both the IS method and the 4-probe method on cylindrical samples cut from the middle of sintered ceramics. Thus, the influence of the surface layer enriched with lanthanum oxide was leveled.
Did the authors check the surface of La2H2O7 samples with XRD (before grinding)?
Reply: No. Only powders were studied by X-ray diffraction.
- Is the sentence “Note the slight excess of lanthanum on the surface of the La2(Hf2-xLax)O7-x/2 (x = 0.1) stuffed pyrochlore compared to La2Hf2O7 (Table 2)” correct? Table 2 shows more lanthanum on the surface of La2Hf2O7.
Reply: Thank you.
The correct sentence is " Note the excess of lanthanum on the outer surface of the La2Hf2O7 compared to La2(Hf1.9La0.1)O6.95 "stuffed" pyrochlore (Table 2)".
- The authors show TG results and state that “Thus, the second heating showed that, during cooling in an oxygen atmosphere, the material regained most of the physically and structurally bound water.” and “It is well seen that La2Hf2O7 loses surface water and hydroxyl ions up to 650 ºC and above this temperature there is no weight loss, while La2(Hf2-xLax)O7-x/2 (x = 0.1) loses structurally bound water 283 and protons in the 650 - 1000 °C temperature interval”. The processes of hydration/hydrogenation are usually studied in the atmosphere containing water vapour and/or during the change of atmosphere humidity. Simple loss/increase of the mass on heating/cooling does not prove that it is the “structurally bound water” and the concentration of proton defects that changes.
Reply:
In this work, we compare the behavior of ordered pyrochlore La2Hf2O7, in which there are no vacancies capable of hydration (it is actually a dielectric), and the solid solution in which such vacancies are present. It is obvious that in the first there is no weight loss in the 650-1000ºÐ¡ region, while in the proton conductor La2(Hf2-xLax)O7-x/2 (x = 0.1) there is weight loss in that region.
- The authors compare the values of the conductivity obtained with IS and the 4-probe DC method and conclude “ The 700 °C conductivity of La2(Hf2-xLax)O7-x/2 (x = 0.1) in wet air is ~ 8×10-5 S/cm, in agreement with the two-probe measurement results (Fig. 3).”. It seems to me, there is a mistake. Figure 5 shows that log(sigma x T) at 973 K (1.03 ) at wet atmosphere is approximately -4.17, whereas from Fig. 3 for the same temperature is approximately -1.7. It means about two orders of magnitude difference. Needless to say, I do not understand, why they should be similar. Why the authors do not compare activation energies?
Reply:
Thank you for your comment. In Figure 5, a typo on the abscissa axis was corrected, logσ should have been written instead of logσT. The activation energy values in wet and dry air atmospheres are presented in Table 3.

Reviewer 3 Report
Reviewer carefully reads this manuscript. This manuscript explains proton conduction of La2(Hf2-xLax)O7-x/2 pyrochlore type oxides. Reviewer think any major problems with the topic and the material being covered are not found. The experimental methods are as well.
However, reviewer found some inconsistencies between the figures and the text in the interpretation of the results as follows.
(1) Around line 158, the difference between XRD patterns of La2Hf2O7 and La excess La2Hf2O7 is explained to the observation of superstructure reflection of 111, 311, 331 and 531. However, those peaks are observed in both XRD patterns in Fig.1 (a).
(2) At line 205, the results of EDX analysis are explained that “the La/Hf ratio …… to the intended one”. However, La/Hf ratio of La excess La2Hf2O7 in fracture surface is larger than the nominal stoichiometry over standard error. In addition, does the discussion about two colors indicate that obtained specimens isn’t single phase, unlike the X-ray diffraction results?
(3) In section 3.3, and Fig.3, bulk conductivity is shown. By contrast, In section 3.5, and Fig.5, total conductivity is presented. According to both figures, it is clear that grain boundary resistance in them is much higher than that of bulk resistance. However, In line 290, it is noted that “the 700oC conductivity of La excess La2Hf2O7 in wet ai…. In agreement with the two-probe measurement results.”
(4) In section 3.4, the authors insist that TG curves changes around 650oC. However, it is difficult to recognize any kind of indication of sharpest weight loss between 50 to 650oC. Further, DSC curves are not explained at all. Why the DSC curves changes around 800oC ?
In addition, here are many other areas that are not fully explained. For example, the difference of behavior in bulk and total conductivity in wet/dry air.
As mentioned above, this manuscript includes the point to be improved or clarified. So, reviewer do not recommend to publish this manuscript as presented. However, reviewer think above improvement does not require large revision or additional experiment. Therefore, review conclusion is “minor revision”.
Author Response
Rev.N3. Comments and Suggestions for Authors.
Reviewer carefully reads this manuscript. This manuscript explains proton conduction of La2(Hf2-xLax)O7-x/2 pyrochlore type oxides. Reviewer think any major problems with the topic and the material being covered are not found. The experimental methods are as well.
However, reviewer found some inconsistencies between the figures and the text in the interpretation of the results as follows.
(1) Around line 158, the difference between XRD patterns of La2Hf2O7 and La excess La2Hf2O7 is explained to the observation of superstructure reflection of 111, 311, 331 and 531. However, those peaks are observed in both XRD patterns in Fig.1 (a).
Reply: We agree with this comment. Necessary changes have been made to the text.
(2) At line 205, the results of EDX analysis are explained that “the La/Hf ratio …… to the intended one”. However, La/Hf ratio of La excess La2Hf2O7 in fracture surface is larger than the nominal stoichiometry over standard error. In addition, does the discussion about two colors indicate that obtained specimens isn’t single phase, unlike the X-ray diffraction results?
Reply: Thank you for your comment. It should be noted that X-ray diffraction studies were carried out on powdered materials, and EDX analysis was used to study ceramic samples. Indeed, the average value of the La/Hf ratio on the fracture surface exceeds the nominal stoichiometry within the standard error of the mean (SEM), but it is close to it and significantly less than this ratio on the outer surface. Also, visual information about the inhomogeneity of the color of the samples cannot serve as a basis for considering the samples as polyphasic, however, based on the EDX analysis data and this information, we can conclude that there is a gradient in the La/Hf ratio. This inhomogeneity is apparently associated with a significant inertia of the system, as a result of which the diffusion of atoms proceeds slowly.
The analysis was made on 10 points. It is possible that this number of points is not enough to analyze the composition of La2Hf2O7. This is the first time we have encountered a visually multicolored ceramic sample. It is known that the phase formation of Ln2Hf2O7 (Ln = La, Nd) is slowed down [1]. Obviously, this is due to the inertia of systems with large cations. Diffusion of La2O3 on the surface of ceramic grains, which is related to its basic properties, cannot be ruled out.
Such inertia of La2O3 and especially HfO2 prevents the removal of carbon, and the synthesis process takes more time for traditional solid state synthesis [1].
X-ray diffraction does not see x-ray amorphous La hydroxides and hydroxycarbonates in the small %.
[1]. Anand, V.K., Bera, A.K., Xu, J., Herrmannsdörfer, T., Ritter, C., and Lake, B., Observation of long-range magnetic ordering in pyrohafnate Nd2Hf2O7: a neutron diffraction study, Phys. Rev. B: Condens. Matter Mater. Phys., 2015, vol. 92, paper 184 418. https://doi.org/10.1103/Phys.Rev. B.92.184418
(3) In section 3.3, and Fig.3, bulk conductivity is shown. By contrast, In section 3.5, and Fig.5, total conductivity is presented. According to both figures, it is clear that grain boundary resistance in them is much higher than that of bulk resistance. However, In line 290, it is noted that “the 700oC conductivity of La excess La2Hf2O7 in wet ai…. In agreement with the two-probe measurement results.”
Reply: Thank you for your comment. There was a typo in Figure 5: the abscissa axis caption should read as logσ instead of logσT. Both Figure 3 and Figure 5 present the total conductivity data of the material under study. However, Figure 3 shows a comparative analysis of conductivity for different Ln2(Hf1.9Ln0.1)O6.95 "stuffed" hafnates pyrochlores in dry and wet air. While Figure 5 demonstrates the total conductivity of La2(Hf2–xLax)O7–x/2 (x = 0.1) in different atmospheres. It was not possible to separate the total conductivity into the bulk and grain boundary components in this work.
(4) In section 3.4, the authors insist that TG curves changes around 650oC. However, it is difficult to recognize any kind of indication of sharpest weight loss between 50 to 650oC. Further, DSC curves are not explained at all. Why the DSC curves changes around 800oC ?
Reply: Thank you for your comment. In this work the sharpest weight loss (50-650 oC) for La2Hf2O7 ( Fig.4 b, curve 2) is associated with the release of H2O and CO2. It can be partially assigned to the decomposition of La-containing carbonates and hydroxycarbonates formed due to trapping CO2 from air during cooling and located at grain boundaries or in pores [1, 2]. Earlier [3], we have used mass spectroscopy to study all the effects on the TG and DSC curves during the phase formation of Nd2Hf2O7 from the mechanically activated oxides mixture. It has been found that weight loss between 250 to 650oC is associated with the H2O and CO2 release.
[1]Samuskevich, V.V., Prodan, E.A., and Pavlyuchenko, M.M., Effect of the gas phase on the thermal decomposition of lanthanum carbonate, Zh. Neorg. Khim., 1972, vol. 17, pp. 2067–2071.
[2]Pavlyuchenko, M.M., Samuskevich, V.V., and Prodan, E.A., Effect of water vapor on lanthanum carbonate octahydrate dehydration, Vestn. Akad. Nauk Bel. SSR, Ser. Khim. Nauk, 1970, vol. 6, pp. 11–15.
[3] A.V. Shlyakhtina, G.A. Vorobieva, A.N. Shchegolikhin et al. Phase Relations and Behavior of Carbon-Containing Impurities in Ceramics Prepared from Mechanically Activated Ln2O3 + 2HfO2 (Ln = Nd, Dy) Mixtures. Inorganic Materials, 2020, Vol. 56, No. 5, pp. 528–542.
The exoeffect on DSC curve near 800ºC ( Fig.4 a, curve 1) can be caused by the structure relaxation after the removal of bulk CO32− groups. It should be noted that after the 2nd heating this effect disappears.
In addition, here are many other areas that are not fully explained. For example, the difference of behavior in bulk and total conductivity in wet/dry air.
Reply: Figs. 3 and 5 demonstrate the total conductivity of La2(Hf2–xLax)O7–x/2 (x = 0.1) in dry and wet atmospheres. It was not possible to separate the total conductivity into the bulk and grain boundary components for La2(Hf2–xLax)O7–x/2 (x = 0.1). So, the total conductivity of La2(Hf2-xLax)O7-x/2 (x = 0.1) “stuffed” pyrochlore is equal to the bulk.
As mentioned above, this manuscript includes the point to be improved or clarified. So, reviewer do not recommend to publish this manuscript as presented. However, reviewer think above improvement does not require large revision or additional experiment. Therefore, review conclusion is “minor revision”.

Round 2
Reviewer 2 Report
The authors introduced many corrections which improved the manuscript. There are, however, some points that still are not convincing.
1) Is decreasing the number of self-citation from 17 to 13 sufficient?
2) The author's answer is: " In this work, we compare the behavior of ordered pyrochlore La2Hf2O7, in which there are no vacancies capable of hydration (it is actually a dielectric), and the solid solution in which such vacancies are present. It is obvious that in the first there is no weight loss in the 650-1000ºÐ¡ region, while in the proton conductor La2(Hf2-xLax)O7-x/2 (x = 0.1) there is weight loss in that region."
I do not agree with the phrase "It is obvious.." Nothing in science is obvious as long as it is proven. The compounds containing lanthanum in this temperature range may very well lose CO2. Your experiment was not correct from the point of view of hydration studies, so please, use less arbitrary statements.
3) Indeed, "The activation energy values in wet and dry air atmospheres are presented in Table 3." . Where are they discussed?
Author Response
Comments and Suggestions for Authors
The authors introduced many corrections which improved the manuscript. There are, however, some points that still are not convincing.
1) Is decreasing the number of self-citation from 17 to 13 sufficient?
Reply: Thak you for the comment.
We have changed 3 references in the references list. In detail, we have reduced self-citations (3) and included a new reference of another research group (1).
Further, our references modification may destroy the logic of the manuscript.
2) The author's answer is: " In this work, we compare the behavior of ordered pyrochlore La2Hf2O7, in which there are no vacancies capable of hydration (it is actually a dielectric), and the solid solution in which such vacancies are present. It is obvious that in the first there is no weight loss in the 650-1000ºÐ¡ region, while in the proton conductor La2(Hf2-xLax)O7-x/2 (x = 0.1) there is weight loss in that region."
I do not agree with the phrase "It is obvious.." Nothing in science is obvious as long as it is proven. The compounds containing lanthanum in this temperature range may very well lose CO2. Your experiment was not correct from the point of view of hydration studies, so please, use less arbitrary statements.
Reply: Thank you for the comment.
The electrode contacts for the La2(Hf1.9La0.1)O6.95 sample were made by firing ChemPur C3605 paste with colloidal platinum at 950 °C for 0.5 h. This temperature (950°C) is higher than the temperature of the bulk CO32− groups removal in the powder (~ 800 °C). Therefore, this effect cannot influence conductivity measurements because the electrode samples were preannealed at a higher temperature (950 °C). Please, look at the Fig.4 a, curve 2. You can see that after the 2nd heating this effect absolutly disappears.
3) Indeed, "The activation energy values in wet and dry air atmospheres are presented in Table 3." . Where are they discussed?
Reply: Thank you for the comment. We have included this discussion in the text.
It is obvious (Table 3) that below 800 °C the activation energies for conduction in the Ln2(Hf1.9Ln0.1)O6.95 (Ln = La, Nd) "stuffed" hafnates pyrochlores in wet air is lower than that of dry air. This is usual proton conductorsʹs behaviour. Ln2(Hf1.9Ln0.1)O6.95 (Ln = Sm, Eu) "stuffed" pyrichlores have the same activation energy ( Table 3) for conductivity, which is independent of humidity.
